# Physiological and Transcriptomic Analyses Reveal the Mechanisms Underlying Methyl Jasmonate-Induced Mannitol Stress Resistance in Banana

**DOI:** 10.3390/plants13050712

**Published:** 2024-03-03

**Authors:** Jiaxuan Yu, Lu Tang, Fei Qiao, Juhua Liu, Xinguo Li

**Affiliations:** 1School of Tropical Agriculture and Forest, Hainan University, Haikou 570228, China; 21110710000041@hainanu.edu.cn (J.Y.); tangluya2023@163.com (L.T.); 2National Key Laboratory for Tropical Crop Breeding, Haikou 570228, China; liujuhua@itbb.org.cn; 3Tropical Crops Genetic Resources Institute, Chinese Academy of Tropical Agricultural Sciences, Haikou 571737, China; fei.qiao@catas.cn

**Keywords:** banana, MeJA, RNA-seq, mannitol-induced stress

## Abstract

Exogenous methyl jasmonate (MeJA) application has shown promising effects on plant defense under diverse abiotic stresses. However, the mechanisms underlying MeJA-induced stress resistance in bananas are unclear. Therefore, in this study, we treated banana plants with 100 μM MeJA before inducing osmotic stress using mannitol. Plant phenotype and antioxidant enzyme activity results demonstrated that MeJA improved osmotic stress resistance in banana plants. Thereafter, to explore the molecular mechanisms underlying MeJA-induced osmotic stress resistance in banana seedlings, we conducted high-throughput RNA sequencing (RNA-seq) using leaf and root samples of “Brazilian” banana seedlings treated with MeJA for 0 h and 8 h. RNA-seq analysis showed that MeJA treatment upregulated 1506 (leaf) and 3341 (root) genes and downregulated 1768 (leaf) and 4625 (root) genes. Then, we performed gene ontology and Kyoto Encyclopedia of Genes and Genomes analyses on the differentially expressed genes. We noted that linoleic acid metabolism was enriched in both root and leaf samples, and the genes of this pathway exhibited different expression patterns; *9S-LOX* genes were highly induced by MeJA in the leaves, whereas *13S-LOX* genes were highly induced in the roots. We also identified the promoters of these genes, as the differences in response elements may contribute to tissue-specific gene expression in response to MeJA application in banana seedlings. Overall, the findings of this study provide insights into the mechanisms underlying abiotic stress resistance in banana that may aid in the improvement of banana varieties relying on molecular breeding.

## 1. Introduction

Banana (*Musa* spp.)—one of the most important economic fruits in the global fruit market—is an African staple food [1]. Owing to shallow roots and a permanently green canopy, banana fruit production and quality are negatively affected by drought and salt stress [2,3]. Furthermore, drought, salt, and cold stress can result in water loss, which in turn induces osmotic stress. Therefore, in various banana-growing regions worldwide, drought stress poses a significant challenge, leading to substantial yield losses ranging from 20% to 65%, as observed in the East African Highlands [4]. Furthermore, banana cultivation requires an annual rainfall of approximately 2000–2500 mm, evenly distributed throughout the year, to achieve optimal yield output [5]. Imposing a one-month drought during the flowering stage can thus result in a significant reduction of up to 42% in banana bunch weight [6].

Moreover, stress resistance in banana plants is primarily influenced by their genotype. Banana plants with the AAA genome, such as dessert bananas, exhibit higher sensitivity to drought and abiotic stresses compared to varieties possessing the AAB genome, which display moderate stress tolerance. Nonetheless, cooking banana species with the ABB genome demonstrate the highest level of drought tolerance among all banana types [6]. Therefore, it is necessary to determine the molecular mechanisms underlying stress resistance in banana species to improve their resistance status.

Methyl jasmonate (MeJA), a derivative of jasmonic acid (JA), is a crucial plant growth regulator involved in responses against biotic and abiotic stresses [7,8,9,10] In this regard, the application of exogenous MeJA in enhancing abiotic stress resistance has been studied extensively in plants. For example, a study on wheat (*Triticum aestivum* L.) showed that exogenous MeJA helped avoid stress by improving D1 protein abundance and the efficiency of photosystem II under heat stress [11]. Similarly, in perennial ryegrass (*Loliumperenne* L.), exogenous MeJA induced heat tolerance by maintaining an adequate plant water content, low electrolyte leakage, and an adequate malondialdehyde (MDA) content. The application of MeJA also relieved the impact of vibrational injury in broccoli [12].

Recent studies on the role of MeJA in banana have focused on both biotic and abiotic stresses. *C-repeat Binding Factor* (*CBF*) and *Inducer of CBF Expression* (*ICE*) are upregulated in response to cold stress in various plant species. In banana fruits, MeJA treatment significantly enhanced resistance to cold stress by inducing the expression of *ICE–CBF* cold-responsive pathway genes [13,14]. Furthermore, several transcription factors induced by MeJA application play important roles in cold stress tolerance. For example, MeJA improves the expression of *myelocytomatosis 2a* (*MaMYC2a*) and *2b* (*MaMYC2b*), both of which interact with MaICE1 [12] and protect banana plants from cold stress. In banana, MaICE1 directly interacts with basic helix-loop-helix 1/2/4 (MabHLH1/2/4), and their promoters can be activated by cold stress and MeJA treatment [14]. Additionally, the genes encoding the banana lysyl oxidase (*MaLOX*) family are induced by MeJA and play an important role in resistance to various stresses, such as low- and high-temperature stresses and *Fusarium oxysporum* f. sp. Cubense tropical infection [15]. These studies have revealed some functions of MeJA in banana species; however, the broader role of MeJA in banana species is unknown.

Exogenous application of MeJA has been commonly used to induce abiotic stress resistance in various plant species [16,17,18]. Although a few studies have focused on the application of MeJA to enhance stress resistance in banana, a comprehensive approach to understanding the molecular mechanisms underlying this strategy is far from being explored. Therefore, in this study, to determine the possible molecular mechanisms underlying MeJA-induced abiotic stress resistance in banana, RNA sequencing (RNA-seq) was used to analyze the expression of genes induced upon MeJA application. We found that *LOX* genes induced by MeJA application exhibited tissue-specific expression in the roots and leaves of banana plants, which may have been responsible for enhancing abiotic stress resistance. Hence, we further analyzed the promoters of *LOX* genes to explore the reason for their tissue-specific expression. Thus, this study aimed to provide a theoretical basis for MeJA-induced abiotic stress resistance in bananas.

## 2. Results

### 2.1. MeJA Treatment Improved Abiotic Stress Resistance in Banana

MeJA has been shown to enhance stress resistance in several plants [7,8,9,10,11,12], but its function in abiotic stress tolerance in banana is unclear. Therefore, in this study, we exposed banana seedlings to MeJA to determine its role in stress resistance and used mannitol (Ma) as an osmotic stressor.

Banana seedlings were exogenously treated with 100 μM MeJA for 8 h (denoted as 8 h MeJA), and the untreated ones served as the control (0 h MeJA). Then, MeJA-treated and control seedlings were placed in 1 M Ma solution for 48 h, or not exposed to Ma (denoted as 48 h Ma and 0 h Ma, respectively). Under osmotic stress conditions (simulated using Ma), MeJA-treated banana seedlings exhibited higher stress resistance than the control seedlings. Additionally, the oldest leaves fell off and wilted in the 0 h MeJA + 48 h Ma group; however, the seedlings in the 8 h MeJA + 48 h Ma group did not exhibit observable stress symptoms (Figure 1A).

Under stress conditions, the activities of peroxidase (POD), catalase (CAT), and superoxide dismutase (SOD) and the content of MDA are used to measure the degree of the plant stress. The superoxide anion radical (O^2−^) plays a crucial role as an intermediate product in various physiological and biochemical reactions within biological cells, exhibiting significant accumulation under stress conditions. SOD effectively eliminates O^2−^ through a disproportionation reaction, resulting in the production of H_2_O_2_ and O_2_. Subsequently, CAT or POD facilitate the conversion of H_2_O_2_ into water and O_2_. However, O^2−^ initiates an attack on unsaturated fatty acids, leading to a decrease in membrane fluidity and subsequent disruption of cellular physiological functions. MDA serves as the principal end product resulting from lipid peroxidation. This process represents a commonly observed mechanism underlying oxidative damage at the cellular level. Hence, we measured the activities of POD, CAT, and SOD and the MDA content in the leaves of banana seedlings (Figure 1B). After Ma treatment, the activities of POD and SOD and the content of MDA increased considerably, but MeJA restrained the changes. However, the activity of CAT exhibited a contrasting trend. These results suggested that MeJA reduced the extent of damage in banana seedlings under osmotic stress, thereby improving abiotic stress resistance.

### 2.2. Sample Preparation and RNA Quality Assessment

To elucidate the mechanism underlying MeJA-induced abiotic stress resistance in banana, high-throughput RNA-seq was performed using leaf and root samples from seedlings exposed to 8 h of MeJA treatment (8 h MeJA), and the control group comprised samples not exposed to MeJA (0 h MeJA). Total RNA of each group included RNA samples from three biological replicates (in the sample description of RNA-seq, “A” represents banana leaves and “B” represents banana roots, and biological replicates are named “_1”, “_2”, and “_3”).

To ensure adequate quality and quantity of total RNA, we determined the purity, concentration, and completeness of RNA using agarose gel electrophoresis (Appendix A), a Nanodrop 2100 (Thermo Fisher Scientific, Waltham, MA, USA), Qubit 2.0 (Thermo Fisher Scientific, MA, USA), and an Agilent Bioanalyzer 2100 (Agilent Technologies, Santa Clara, CA, USA) (Appendix A). Thereafter, RNA samples of adequate quality were used for sequencing and cDNA library construction.

Before data analysis, quality control was performed on RNA-seq data. The total number of clean reads per library ranged from 39,442,658 to 55,742,402, and clean reads in each library accounted for >97% of raw reads. A total of 90.67 gb of clean bases were generated in 12 cDNA libraries using an Illumina HiSeq 2500 (Illumina, San Diego, CA, USA) sequencing platform (Appendix A). The base error rate of each library was 0.03%, Q20 values were >97%, and Q30 values were >88% (Appendix A). The GC content ranged from 51.96% to 55.71% (Appendix A).

Following quality control, the clean reads were mapped to the reference banana A genome (*M. acuminata*). On average, 89.98% of the reads from all samples were effectively mapped to different regions within the reference genome (Appendix A). Subsequently, the fragments per kilobase of transcript per million mapped reads (FPKM) value was determined. Approximately 34% of the expressed genes were in the FPKM range of 0–1, 14% in the FPKM range of 1–3, 29% in the FPKM range of 3–15, 17% in the FPKM range of 15–60, and 6% in the FPKM range >60 (Appendix A). Pearson’s correlation analysis showed a high correlation (R = 0.950–0.991) between the biological replicates of each group (Figure 2A), indicating that the transcriptome data used in this study were of high quality.

### 2.3. Identification and Analysis of Differentially Expressed Genes

To identify the effect of MeJA on banana leaves or roots, we screened differentially expressed genes (DEGs) with a |log_2_ fold-change (FC)| > 1 and a false discovery rate < 0.05 in each pairwise comparison (leaves: 0 h vs. 8 h after treatment; roots: 0 h vs. 8 h after treatment). In the leaves of seedlings treated with MeJA, 3274 genes were differentially expressed, including 1506 upregulated and 1768 downregulated genes. In contrast, we identified 3341 upregulated and 4625 downregulated genes in the roots of seedlings treated with MeJA, indicating a total of 7966 DEGs in the roots. Figure 2B illustrates the volcano plots of the distribution of different multiples and the significance of DEGs in the leaves and roots of MeJA-treated seedlings. Based on the plots, we found that MeJA exhibited dramatic effects on banana seedlings. Figure 2C shows a Venn diagram obtained from the analysis of DEGs and suggests that 1504 genes were differentially expressed in both the leaves and roots of MeJA-treated seedlings; however, 1770 genes were differentially expressed only in the leaves and 6462 genes were differentially expressed only in the roots.

### 2.4. Gene Ontology Enrichment Analysis of the DEGs

Gene ontology (GO) enrichment analysis of DEGs was used to gain insights into the functions of DEGs based on three terms, including biological process, molecular function, and cellular component. A total of 162 significant GO terms were enriched in the upregulated genes of banana leaves, with 100 GO terms enriched in the biological process category, 34 GO terms enriched in the cellular component category, and 28 GO terms enriched in the molecular function category. We selected the top 30 functional categories, including 18 biological process GO terms, 6 cellular component GO terms, and 6 molecular function GO terms, to plot a GO enrichment bar chart (Figure 3A). In the biological process category, the major categories were “jasmonic acid metabolic process” (GO: 0009694), “fatty acid beta-oxidation” (GO: 0006635), and “lipid oxidation” (GO: 0034440). The DEGs in the cellular component category were concentrated in the “glyoxysome” (GO: 0009514), “obsolete integral component of peroxisomal membrane” (GO: 0005779), and “obsolete intrinsic component of peroxisomal membrane” (GO: 0031231). In the molecular function category, “linoleate 13S-lipoxygenase activity” (GO: 0016165), “chitinase activity” (GO:0004568), and “alpha-amylase inhibitor activity” (GO:0015066) were the most enriched.

A total of 116 GO terms were enriched in the downregulated genes of banana leaves, including 8 GO terms enriched in the molecular function category, 42 GO terms enriched in the cellular component category, and 66 GO terms enriched in the biological process category. The top 30 functional categories were selected to plot the GO enrichment bar chart, which included important GO terms (Figure 3B). In the biological process category, the major categories were “photosynthesis, light harvesting in photosystem I” (GO: 0009768), “photosynthesis, light harvesting” (GO: 0009765), and “photosynthesis, light reaction” (GO:0019684). The DEGs in the cellular component category were concentrated in the “photosystem” (GO: 0009521), “photosystem II” (GO: 0009523), and “chloroplast thylakoid membrane protein complex” (GO: 0098807). In the molecular function category, “pigment binding” (GO: 0031409), “chlorophyll binding” (GO:0016168), and “protein domain specific binding” (GO:0019904) were the most enriched.

A total of 211 GO terms were enriched in the upregulated genes of banana roots, including 31 GO terms enriched in the molecular function category, 15 GO terms enriched in the cellular component category, and 165 GO terms enriched in the biological process category. The top 30 functional categories were selected to plot the GO enrichment bar chart, which included important GO terms (Figure 3C). In the biological process category, the major categories were “response to alkaline pH” (GO: 0010446), “cellular response to alkaline pH” (GO: 0071469), and “regulation of cellular response to alkaline pH” (GO: 1900067). The DEGs in the molecular function category were concentrated in the “galactolipase activity” (GO: 0047714), “phospholipase A1 activity” (GO: 0008970), and “triglyceride lipase activity” (GO: 0004806).

A total of 114 GO terms were enriched in the downregulated genes of banana roots, including 22 GO terms enriched in the molecular function category, 18 GO terms enriched in the cellular component category, and 74 GO terms enriched in the biological process category. The top 30 functional categories were selected to plot the GO enrichment bar chart, which included important GO terms (Figure 3D). In the biological process category, the major categories were “snRNA pseudouridine synthesis” (GO: 0031120), “snRNA modification” (GO: 0040031), and “DNA replication initiation” (GO: 0006270). The DEGs in the cellular component category were concentrated in the “box H/ACA snoRNP complex” (GO: 0031429), “box H/ACA RNP complex” (GO: 0072588), and “filiform apparatus” (GO: 0043680). In the molecular function category, “xyloglucan: xyloglucosyl transferase activity” (GO: 0016762), and “amino acid transmembrane transporter activity” (GO: 0015171) were the most enriched.

### 2.5. Kyoto Encyclopedia of Genes and Genomes Enrichment Analysis of DEGs

Kyoto Encyclopedia of Genes and Genomes (KEGG) enrichment analysis of the DEGs was used to determine the primary biological pathways induced upon MeJA treatment in banana leaves and roots, and the top 20 pathways were selected for further analysis (Figure 4A,B). Enriched factors were compared to identify the major pathways that were affected by MeJA treatment. In the leaves, “photosynthesis-antenna proteins” (mus00196), “photosynthesis” (mus00195), and “linoleic acid metabolism” (mus00591) were enriched, indicating that the DEGs accounted for a high proportion of these pathways (Figure 4A). However, the top three pathways exhibiting highly enriched factors in the roots were “linoleic acid metabolism” (mus00591), “selenocompound metabolism” (mus00450), and “biotin metabolism” (mus00780) (Figure 4B).

These results suggested that photosynthesis was altered in the seedlings to adapt to harsh environments and improve stress resistance. As banana roots do not perform photosynthesis, they respond to abiotic stress by regulating superincumbent and biotin metabolism. Notably, several DEGs of metabolic pathways and secondary metabolite biosynthesis pathways were enriched in banana leaves and roots at low levels, and the linoleic acid metabolism always exhibited high enrichment that was reduced upon MeJA treatment.

### 2.6. Regulatory Mechanisms in Leaves Altered upon MeJA Treatment

KEGG analysis of leaf samples revealed that the genes involved in photosynthesis-related pathways were enriched in the leaves, and their expression levels are shown in Figure 5. The genes involved in “photosynthesis” (mu00195) (Figure 5A) and “photosynthesis-antenna proteins” (mu00196) (Figure 5B) were highly downregulated after MeJA treatment. Although a few genes (*Ma09_g30430* and *Ma04_g38200*) in the photosynthetic electron transport chain were upregulated, these observations indicated that MeJA altered photosynthesis by influencing the photoelectron transport chain and photosynthetic efficiency, which may have resulted in MeJA-induced abiotic stress resistance in banana leaves.

### 2.7. Regulatory Mechanisms in Roots Altered upon MeJA Treatment

The “selenocompound metabolism” and “biotin metabolism” pathways were enriched in banana roots. Additionally, we identified the following DEGs involved in Se hyperaccumulation and “selenocompound metabolism” (Figure 6A): *3′*-*phosphoadenosine 5′*-*phosphosulfate synthase* (*PAPSS*), *thioredoxin reductase NTRC* (*Txnrd*), *cysteine desulfurase*, *selenocysteine lyase* (*CpNitS*), *cystathionine gamma-synthase* (*CGS*), *cysteine-S-conjugate beta-lyase* (*CBL*), *5-methyltetrahydropteroyltriglutamate-homocysteine methyltransferase* (*MET*), and *methionyl-tRNA synthetase* (*MetRS*). *Txnrd*, *CBL*, and *MetRS* were downregulated upon MeJA treatment, but *CpNifS*, and *CGS* were upregulated. We also noted up- and downregulation in *PAPSS* and *MET* (Figure 6A).

In the “biotin metabolism” pathway, the following eight genes were enriched: *FabF*, *FabG*, *FabZ*, *FabI*, *BioF*, *BioA*, *BioD*, and *BioB* (Figure 6B). Biotin biosynthesis is a conserved four-step pathway. In the first step, 7-keto-8-aminopelargonic acid (KAPA) synthase catalyzes the synthesis of KAPA in peroxisomes. The intermediate two-step reaction is catalyzed by the bifunctional BIO3-BIO1 enzyme [19,20]. The final reaction involves the conversion of desthiobiotin to biotin, which is catalyzed by a biotin synthase [21,22,23].

We noted that the expression of genes involved at the beginning of biotin synthesis was altered upon MeJA treatment. For example, *FabF* (encoding 3-oxoacyl-[acyl-carrier-protein] synthase II), *FabZ* (encoding 3-hydroxyacyl-[acyl-carrier-protein] dehydratase), and *FabI* (encoding enoyl-[acyl-carrier protein] reductase I) were downregulated upon MeJA treatment. *FabG* (encoding 3-oxoacyl-[acyl-carrier protein] reductase) and most other genes were upregulated upon MeJA treatment.

Additionally, the genes directly involved in biotin synthesis were affected by MeJA treatment. For example, *BioF* (encoding 8-amino-7-oxononanoate synthase) was upregulated, whereas *BioA* (encoding 7,8-diaminononanoate synthase) was downregulated in banana roots. Dethiobiotin synthetase encoded by *BioD* also exhibited altered expression levels. The biotin synthase encoded by *BioB*, which affects the end products of the pathway, was upregulated.

### 2.8. Linoleic Acid Metabolism Showed Tissue-Specific Enrichment

The “linoleic acid metabolism” was enriched upon MeJA treatment in the leaves and roots of banana seedlings and exhibited significant effects on abiotic stress resistance in banana. The heatmap shown in Figure 7 reveals the diversity of linoleic acid metabolism changes after MeJA treatment between the leaves and roots of banana seedlings (Figure 7).

Lecithin synthesis involves the synthesis of linoleate, which is catalyzed by secretory phospholipase A2 (PLA2). Thereafter, linoleate synthesizes 13(S)-HODE and 9(S)-HODE by reacting with 13S-LOX and 9S-LOX, respectively. We noted that *13S-LOX* genes and *9S-LOX* genes were upregulated in both leaves and roots. However, the expression levels of *9S-LOX* genes increased more significantly in the leaves, whereas the expression levels of *13S-LOX* genes increased more remarkably in the roots. Notably, *PLA2* exhibited distinct changes in leaves or roots. The expression level of *Ma07_g11670* was higher in the leaves than in the roots. *Ma10_g25490* was considerably downregulated in the roots but exhibited no change in the leaves. *Ma08_g14590* was remarkably upregulated in the leaves but downregulated in the roots. The expression level of *Ma03_g15580* in the roots was higher than that in the leaves.

### 2.9. Analysis of the Promoter Region of the Genes Involved in Linoleic Acid Metabolism

The genes involved in linoleic acid metabolism exhibited tissue-specific expression upon MeJA treatment. Hence, it was necessary to explore the factors affecting the expression of these genes. The promoter sequences of these genes were retrieved from the banana genome database (https://banana-genome-hub.southgreen.fr/, accessed on 12 March 2023) (Appendix A), and the *cis*-acting elements (Appendix A) and transcription factor (TF)-binding motifs (Appendix A) were analyzed using PlantRegMap (http://plantregmap.gao-lab.org/binding_site_Prediction_result.php, accessed on 21 March 2023) and PlantPAN 3.0 (http://plantpan.itps.ncku.edu.tw/, accessed on 23 March 2023).

The *cis*-acting element analysis showed that there were many light-response elements in the promoter region of *13S-LOX* genes (Figure 8A), including AT1-motif, Box II, P-box, ACE, and Sp1 elements. These results explained the specific expression of *13S-LOX* genes in the leaves. Additionally, *13S-LOX* and *9S-LOX* genes contained *cis*-acting regulatory elements involved in MeJA responsiveness (ABRE, TGACG, and CGTCA motifs), elucidating why banana roots and leaves exhibited altered expression levels of *13S-LOX* and *9S-LOX* genes in response to MeJA treatment. Moreover, the TF-binding motifs in the promoters included MYB, bHLH, WRKY, NAC, and other TF family-binding sites (Figure 8B).

### 2.10. Verification of the Expression Patterns of DEGs

To validate the RNA-seq results, 16 DEGs (4 upregulated and 4 downregulated in the leaves, and 4 upregulated and 4 downregulated in the roots) were selected for real-time quantitative polymerase chain reaction (RT-qPCR) analysis (Figure 9). These genes included seven TF-coding genes (*TIFY9-complete*, *TIFY9C*; *MYC4-like*, *MYC4L*; *ethylene-responsive transcription factor 1B*, *ERF1B*; *calcium-dependent protein kinase isoform 2*, *CDPK2*; *bZIP transcription factor*, *bZIP*; *transcription factor bHLH137-like*, *bHLH137L*; *transcription factor TGA2-like*, *TGA2L*; *WRKY transcription factor 13*, *WRKY13*), four genes related to secondary metabolism (*allene oxide synthase*, *AOS*; *1-aminocyclopropane-1-carboxylate synthase 2*, *ACS2*; *9-cis-epoxycarotenoid dioxygenase 1*, *NCED2*; *beta-amylase 3*, *BAM3*), four phytohormone receptor genes (*zeaxanthin epoxidase*, *ZEP*; *abscisic acid receptor PYL8-like*, *PYL8*; *auxin transporter-like protein 2*, *AUX2*), and an uncharacterized gene. We analyzed the FC of FPKM and RT-qPCR data after log2 processing and found that despite minor differences in expression levels indicated by RT-qPCR results, similar trends were noted in both the RNA-seq and RT-qPCR data.

## 3. Discussion

Several studies have shown that MeJA can enhance abiotic stress resistance in plants, and the present study also obtained similar results. We analyzed phenotype observations and conducted physiological and biochemical evaluations, and noted that MeJA reduced the degree of damage caused by Ma-induced osmotic stress. In certain studies, MeJA has been found to mitigate stress damage by enhancing the activity of SOD, POD, and CAT. A previous investigation demonstrated that pretreating grains with 0.05 mM MeJA effectively alleviated drought stress in maize plants and significantly increased the levels of total carbohydrates, total soluble sugar, polysaccharides, free amino acids, total proline, and proteins. Additionally, MeJA enhanced the activities of CAT, POD, and SOD while elevating indole acetic acid content [24]. Furthermore, MeJA treatment enhanced drought resistance in cauliflower (*Brassica oleracea* L.) seedlings through activation of enzymatic systems, such as SOD, POD, and CAT, along with non-enzymatic antioxidant systems, including proline and soluble sugar accumulation. MeJA also promoted chlorophyll accumulation, net photosynthetic rate, leaf relative water content, and endogenous abscisic acid level while suppressing lipid peroxidation [25].

However, studies have indicated different ways by which MeJA improves plant stress resistance. The presence of 0.1 mM MeJA exerted a significant impact on cellular growth and flavonoid production in *Hypericum perforatum* by downregulating CAT activity and simultaneously upregulating phenylalanine ammonia-lyase activity [26]. MeJA also enhanced the recovery of salinity-stressed rice plants by modulating the balance of abscisic acid (ABA) and alleviating the inhibitory impact of salt stress on photosynthetic rate [27]. MeJA treatment on root-irrigated soybean seedlings demonstrated that MeJA mitigated the adverse effects induced by NaCl on the chlorophyll content, leaf photosynthetic rate, and leaf transpiration rate through augmentation of proline content and ABA levels [28]. Additionally, a pot experiment performed by soaking seeds in varying concentrations of MeJA revealed that under salt stress conditions, MeJA elevated total soluble proteins, proline accumulation, total soluble sugars, and relative water content, thereby increasing the number of chlorophyll molecules, stomatal conductance, and net photosynthetic rate in *Vigna unguiculata* L. seedlings [29]. These studies suggest that the modulation of plant resistance by MeJA is contingent upon the specific plant variety and the type and intensity of stress imposed on it.

To elucidate the stress resistance mechanism, we used RNA-seq data to identify DEGs after MeJA treatment. KEGG enrichment analysis showed enrichment in “photosynthesis-antenna proteins”, “photosynthesis”, “selenocompound metabolism”, “biotin metabolism”, and “linoleic acid metabolism”. Furthermore, there were more DEGs in the leaves compared to the roots after MeJA treatment. These results revealed that the abovementioned pathways contributed considerably to Ma-induced stress resistance in banana seedlings.

To adapt and survive in a changing environment, plants have evolved sophisticated defense mechanisms. Their direct response to environmental changes is to alter the physiology of photosynthesis. Abiotic stresses (drought, salinity, extreme temperatures, high solar radiation, or a combination) result in excessive light exposure to chloroplasts, thus potentially causing photo-inhibitory damage and photo-oxidative stress to the photosynthetic apparatus [30,31,32]. A previous study showed that MeJA exhibited negative effects on the genes involved in photosynthesis [11]. In *Arabidopsis*, similar results were obtained using proteomics data, that is, photosynthesis and carbohydrate anabolism were reduced after MeJA treatment [33]. This is consistent with the results obtained in this study, that is, after MeJA treatment, most photosynthesis-related genes were downregulated (Figure 5), suggesting that MeJA could improve resistance to environmental stressors by altering photosynthesis.

Selenium is a beneficial element that promotes plant growth and relieves abiotic stress, especially by inducing resistance against metal toxicity and drought stress [34,35]. However, in this study, “selenocompound metabolism” was only enriched in banana roots. Additionally, MeJA induced the expression of *CpNifS*, *MET*, and *MGL*, which produce hydrogen selenide (Se^0^), L-selenomethionine, and methylcellulose, respectively (Figure 6A). Nonetheless, the relationship between these selenocompounds and abiotic stress is unknown and needs further investigation.

Biotin (vitamin B7 or H) is one of the water-soluble B vitamins; it activates enzymes, but it is only synthesized by bacteria, fungi, and plants. Biotin, pyridoxine, riboflavin, and thiamin mutants have been studied extensively in model plants, such as *Arabidopsis*, rice, maize, and tomato [36,37,38,39,40,41], and have shown that B vitamins have surprising emerging roles in plant development, pathogen resistance, and stress tolerance [42,43]. A previous study showed that B vitamins and cofactors were unstable in response to stress when plants sensed changes in their environments [44,45,46,47]. However, supplementing stressed plants with the deficient vitamin(s) improved their stress resistance [38]. For example, in *Arabidopsis thaliana* seedlings, salt stress increased the expression of the biotin synthase gene *AtBIO2*, and the overexpression of *AtBIO2* enhanced biotin content and tolerance to carbonate stress [48]. This partly explains why the genes involved in biotin metabolism were irregularly and violently altered in banana roots after MeJA treatment. However, the enzymes encoded by *BioF* and *BioB*, which act at the first and last steps of biotin synthesis, respectively, were upregulated (Figure 6B).

The expression levels of DEGs in “linoleic acid metabolism” were more similar compared to the genes in “selenocompound metabolism” and “biotin metabolism”. Linoleic acid helps plants develop resilience to adversity and vulnerability to stress, as reported in oil crops such as sunflower (*Helianthus annuus* L.) and canola (*Brassica napus* L.) [49,50]. However, the function of linoleic acid is not clear in many plants, including banana. To improve the understanding of their role in banana, emphasis must be laid on the concept of phyto-oxylipins, which are metabolites produced by the oxidative transformation of unsaturated fatty acids via a series of diverging metabolic pathways. Depending on the specificity of the LOXs involved, these fatty acids can be transformed into 9- or 13-hydroperoxide derivatives, which play an important role against biotic and abiotic stress in plants [51,52,53]. The available evidence suggests that MeJA treatment upregulated the expression of *13S-LOX* and *9S-LOX* genes to increase the content of phyto-oxylipins and stress resistance in banana (Figure 7).

The expression pattern and some functions of *LOX*s have been revealed in banana species by comparing different varieties [54], which were also revealed in this study. However, *13S-LOX* and *9S-LOX* genes indicated considerable tissue specificity, which has not been reported in previous studies. This could be attributed to substrate specificity in different tissues. Several studies have shown that *LOX* genes are induced upon MeJA treatment in many plants. In tea plants, *CsLOX1* is induced under mechanical wounding or MeJA treatment [55]. These reports thus suggest that *LOX* genes are induced by MeJA treatment and abiotic stress, similar to the findings of this study.

## 4. Materials and Methods

### 4.1. Plant Materials and Treatments

“Brazilian” banana (*Musa acuminata* L. AAA group, cv. Brazilian) seedlings were obtained using tissue culture in a culture chamber in the Seed and Seedling Tissue Culture Center, Chinese Academy of Tropical Agricultural Sciences, China. Thereafter, the seedlings were grown in greenhouses after habituation and used for experiments when they had grown to at least 30 cm in length.

For functional verification, the following four treatments were employed: no treatment (control; 0 h MeJA + 0 h Ma); seedlings treated with 100 μM MeJA for 8 h (8 h MeJA + 0 h Ma); seedlings treated with 1 M Ma for 48 h without MeJA treatment (0 h MeJA + 48 h Ma); seedlings treated with 100 μM MeJA for 8 h and 1 M Ma for 48 h (8 h MeJA + 48 h Ma). The activities of CAT, POD, and SOD and the MDA content were measured at specific time points. The four groups of banana seedlings were treated simultaneously, and the tests were conducted using three biological replicates, with each replicate containing three seedlings.

For RNA-seq, banana seedlings were treated with 100 μM MeJA for 0 h and 8 h. After MeJA treatment, the roots and leaves were harvested separately. Each biological replicate contained samples from three seedlings and was frozen immediately in liquid nitrogen and stored at −80 °C until further use.

### 4.2. Measurement of CAT, SOD, and POD Activity and MDA Content

Approximately 2 g of leaves was ground in 5 mL of phosphate-buffered saline (pH = 6.0–6.8). Then, the mixture was spun for 2 min and subjected to centrifugation at 12,000 rpm for 10 min at 4 °C. The supernatant was stored at −20 °C. The activities of the antioxidant enzymes CAT, POD, and SOD were measured using spectrophotometry [56]. MDA content was measured in the cell lysate using a commercial assay kit (Nanjing Jiancheng Bioengineering Institute, Nanjing, China) according to the manufacturer’s instructions.

### 4.3. RNA-Seq and Expression Analysis

For RNA-seq, library preparation and quantification of gene expression levels were conducted at Tianjin Novogene Bioinformatics Technology Co., Ltd. (Tianjin, China). *Musa acuminata* DH Pahang (version 2) genome was used as the reference genome. The reference genome and gene model annotation files were downloaded from the banana genome website (https://banana-genome-hub.southgreen.fr/, accessed on 14 July 2021) directly.

Differential expression analysis (three biological replicates per group) was performed using the R package DESeq (1.18.0), which contained statistical routines for determining differential expression using digital gene expression data with a model based on the negative binomial distribution. The resulting *p* values were adjusted using Benjamini and Hochberg’s approach for controlling the false discovery rate. Genes with an adjusted *p* value < 0.05 were considered DEGs.

### 4.4. GO and KEGG Enrichment Analysis of DEGs

GO enrichment analysis of DEGs was performed using the R package GOseq, wherein gene length bias was corrected. GO terms with corrected *p* values < 0.05 were considered significantly enriched. We used KOBAS 2.0 software to verify the statistical enrichment of DEGs in the KEGG pathways. The expression levels of DEGs are shown as heatmaps generated using the R package pheatmap version 1.0.12.

### 4.5. Promoter Sequence Analysis

The promoter sequences of genes were identified using the banana genome website (https://banana-genome-hub.southgreen.fr/, accessed on 12 March 2023). PlantCARE tool was used to analyze *cis*-acting regulatory elements, and elements with no clear function were filtered out. The TF-binding site prediction tool PlantRegMap was used in combination with *M. acuminata* TF data available at the Center for Bioinformatics, Peking University, to identify TF-binding sites in the promoter regions of the genes.

## 5. Conclusions

The physiological indices of stress resistance and plant phenotype demonstrated that MeJA effectively mitigated the detrimental effects of Ma on banana seedlings. RNA-seq results revealed that MeJA induced significant alterations in the expression of genes associated with “photosynthesis-antenna proteins” (mu00196), “photosynthesis” (mu00195), “selenocompound metabolism” (mus00450), “biotin metabolism pathway” (mus00780), and “linoleic acid metabolism pathway” (mus00591). Notably, KEGG enrichment analysis indicated that the “linoleic acid metabolism pathway” (mus00591) was particularly enriched in both roots and leaves, albeit with distinct expression patterns specific to each tissue. These findings strongly suggest that the “linoleic acid metabolism pathway” is a key mechanism through which MeJA enhances resistance against Ma-induced abiotic stress in banana. The RT-qPCR results further supported these findings. Overall, this study provides a solid theoretical foundation for improving osmotic stress resistance in banana.

## Figures and Tables

**Figure 1 plants-13-00712-f001:**
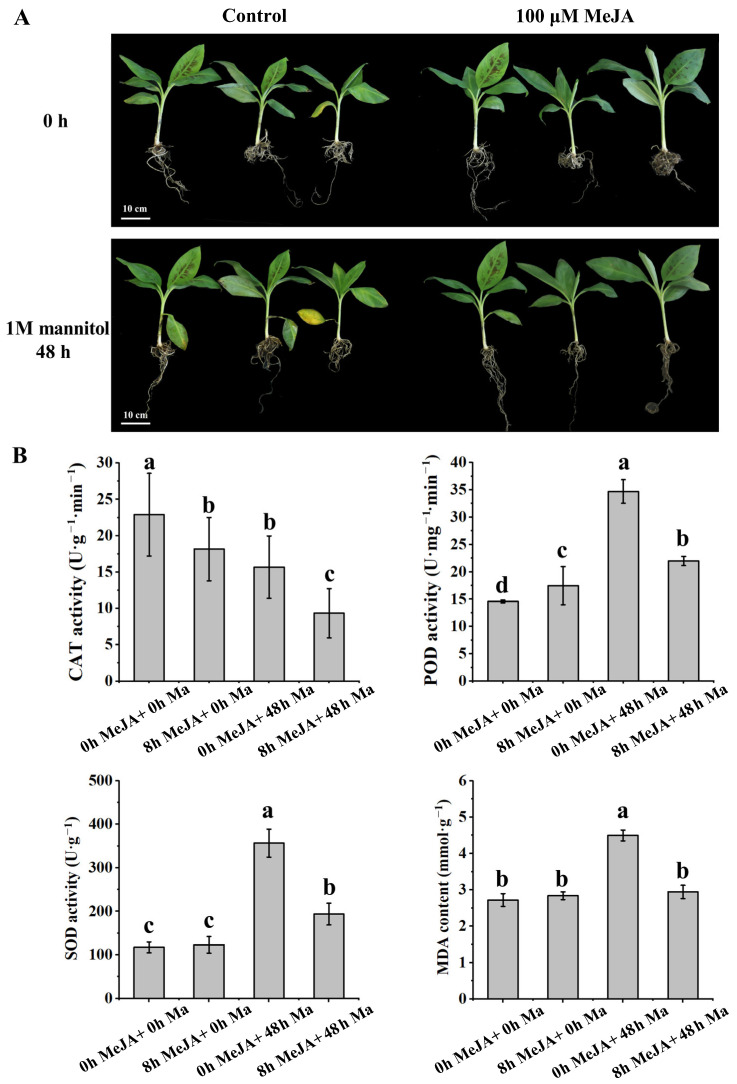
MeJA enhances abiotic stress resistance in banana seedlings. (**A**) Banana seedlings treated with Ma and MeJA. (**B**) Activities of CAT, POD, and SOD, and MDA content. Values are mean ± SD of three independent experiments (*n* = 3). Different lowercase letters above columns indicate statistical differences at *p*  <  0.05. Duncan’s multiple comparison in IBM SPSS Statistics 25 was used for statistical analysis.

**Figure 2 plants-13-00712-f002:**
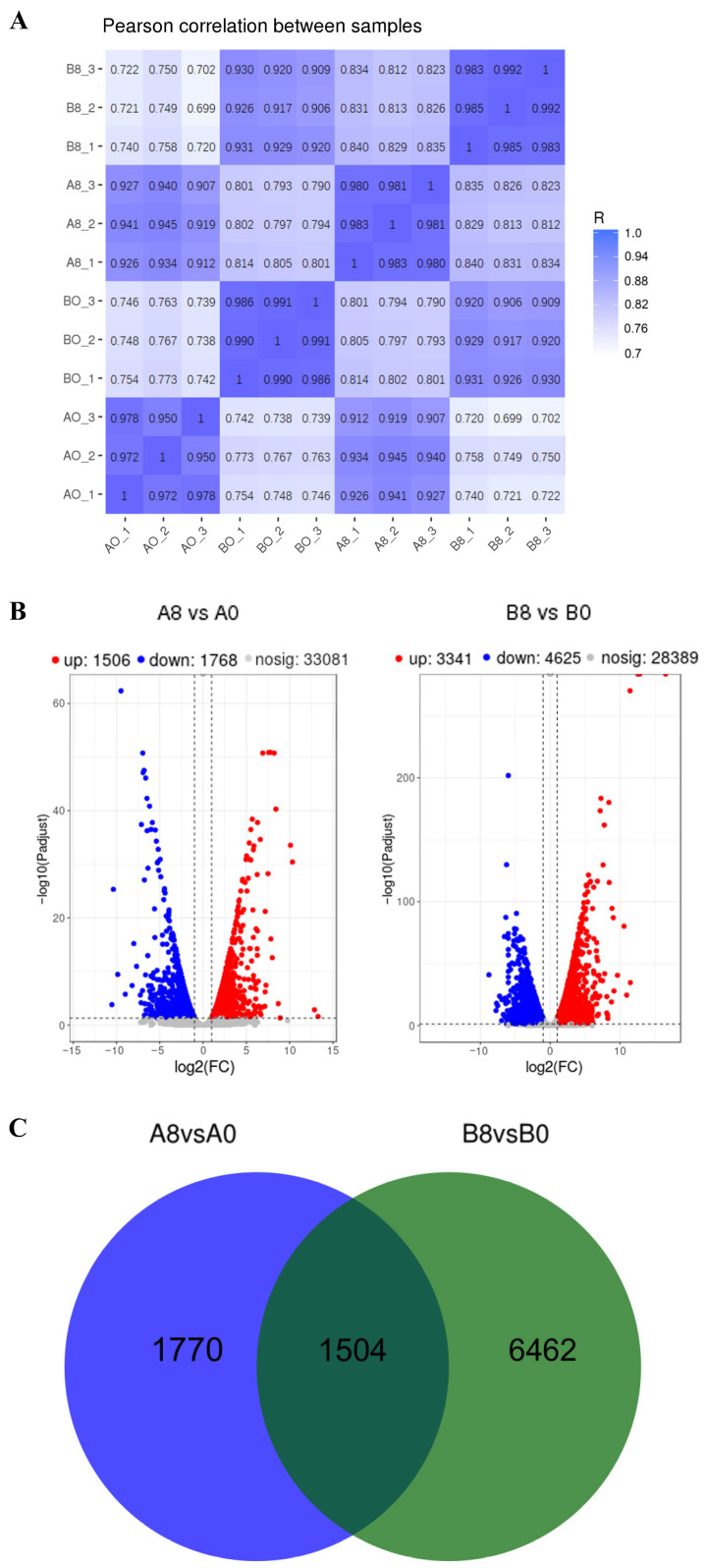
Analysis of transcriptome data. A0, banana leaves exposed to MeJA for 0 h; A8, banana leaves exposed to MeJA for 8 h; B0, banana roots exposed to MeJA for 0 h; B8, banana roots exposed to MeJA for 8 h. (**A**) Pearson’s correlation analysis revealed correlations between the biological replicates. The correlation coefficient R is shown on the heatmap. (**B**) Volcano plots of all genes; green dots represent downregulated genes, and red dots represent upregulated genes. (**C**) Venn diagram of DEGs in MeJA-treated leaf and root samples, as well as control samples.

**Figure 3 plants-13-00712-f003:**
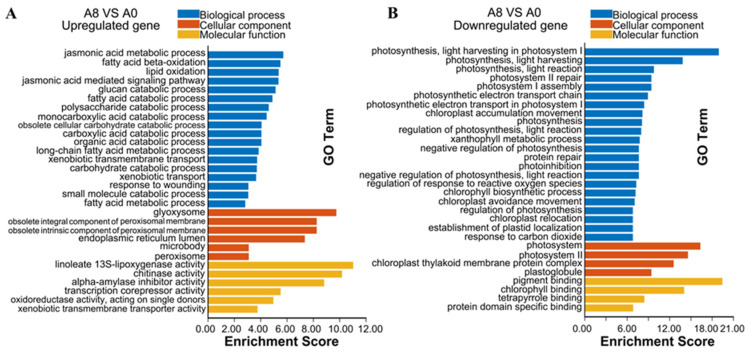
GO enrichment analysis of DEGs in the leaves (**A**,**B**) and roots (**C**,**D**) of banana seedlings. The top 30 terms are shown in the figure. (**A**) Upregulated DEG enrichment in banana leaves. A0, banana leaves exposed to MeJA for 0 h; A8, banana leaves exposed to MeJA for 8 h. (**B**) Downregulated DEG enrichment in banana leaves. (**C**) Upregulated DEG enrichment in banana roots. B0, banana roots exposed to MeJA for 0 h; B8, banana roots exposed to MeJA for 8 h. (**D**) Downregulated DEG enrichment in banana roots. The vertical coordinate denotes GO terms, and the horizontal coordinate denotes enrichment scores. Blue bars represent “biological process”, orange bars represent “cellular component”, and yellow bars represent “molecular function”.

**Figure 4 plants-13-00712-f004:**
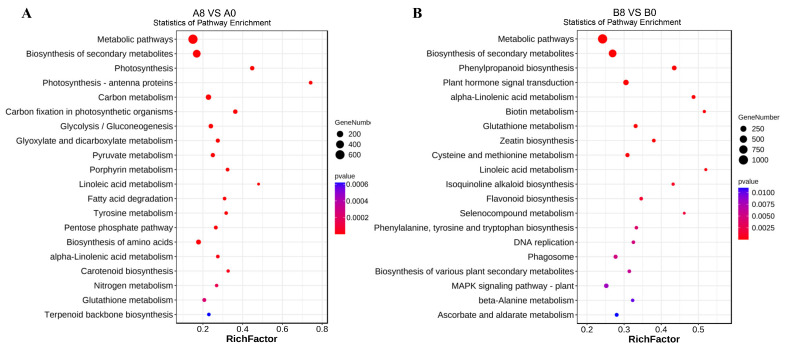
KEGG enrichment analysis of DEGs in the leaves (**A**) and roots (**B**) of banana seedlings. A0, banana leaves exposed to MeJA for 0 h; A8, banana leaves exposed to MeJA for 8 h; B0, banana roots exposed to MeJA for 0 h; B8, banana roots exposed to MeJA for 8 h. The sizes of the dots represent the number of enriched DEGs. The vertical coordinate denotes the KEGG pathways, and the horizontal coordinate denotes the enrichment factor, which reflects the ratio of DEGs to all the genes in the pathway.

**Figure 5 plants-13-00712-f005:**
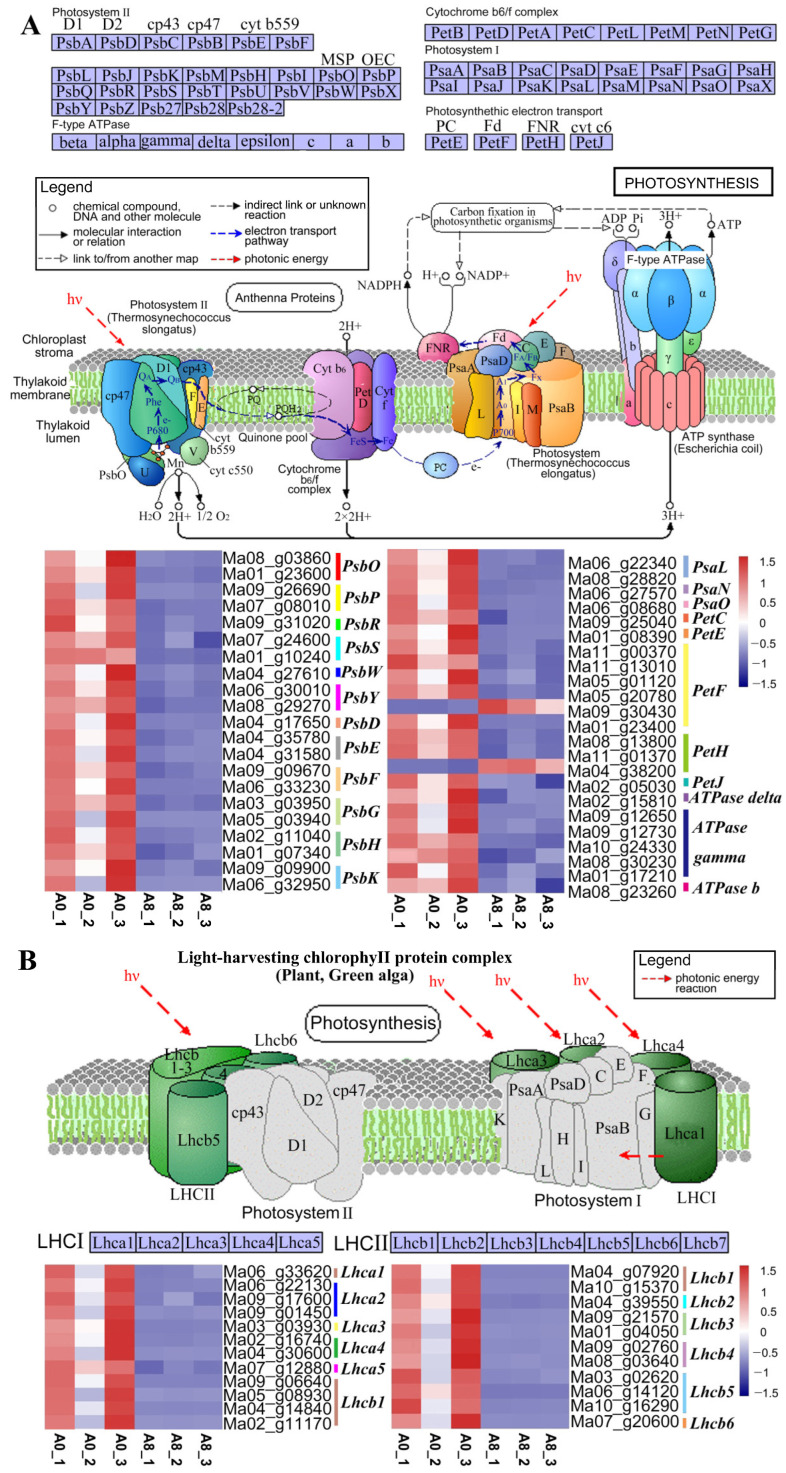
A schematic diagram and heatmap of DEGs involved in the top two enriched pathways in banana leaves. (**A**) “Photosynthesis” pathway and the expression profile of DEGs. (**B**) “Photosynthesis-antenna protein” pathway and the expression profile of DEGs. Red indicates high expression, and blue indicates low expression. A0, banana leaves exposed to MeJA for 0 h; A8, banana leaves exposed to MeJA for 8 h.

**Figure 6 plants-13-00712-f006:**
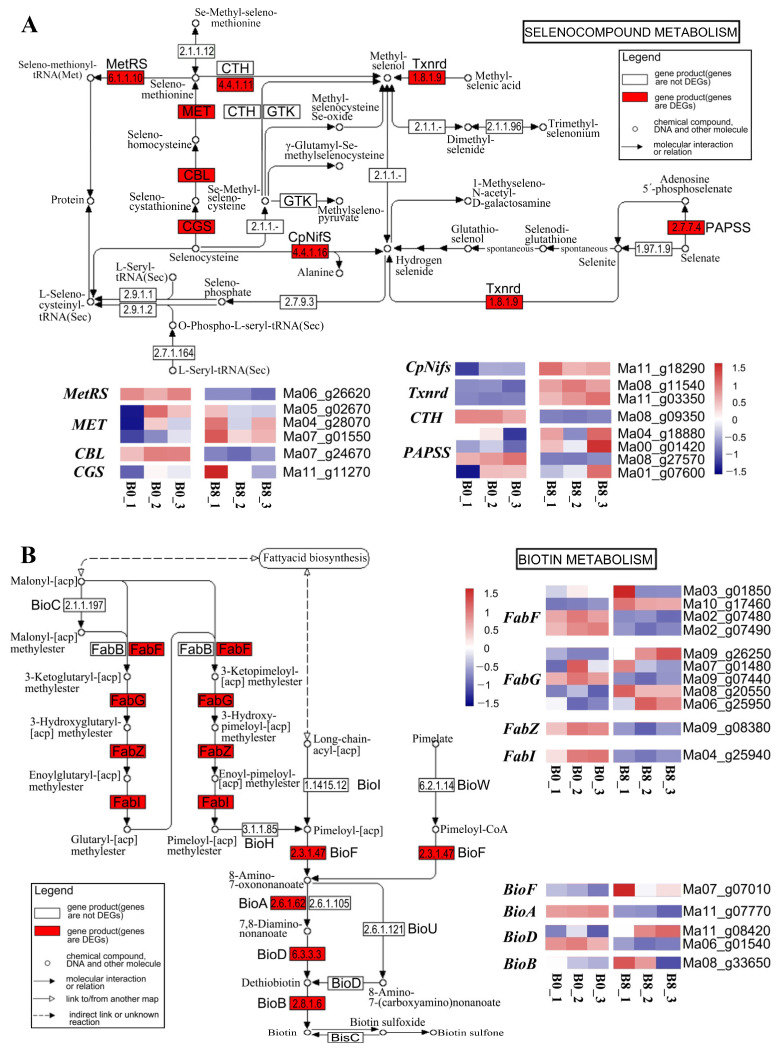
A schematic diagram and heatmap of DEGs involved in the top two enriched pathways in banana roots. (**A**) “Selenocompound metabolism” and the expression profile of DEGs. (**B**) “Biotin metabolism” and the expression profile of DEGs. Red indicates high expression, and blue indicates low expression. B0, banana roots exposed to MeJA for 0 h; B8, banana roots exposed to MeJA for 8 h.

**Figure 7 plants-13-00712-f007:**
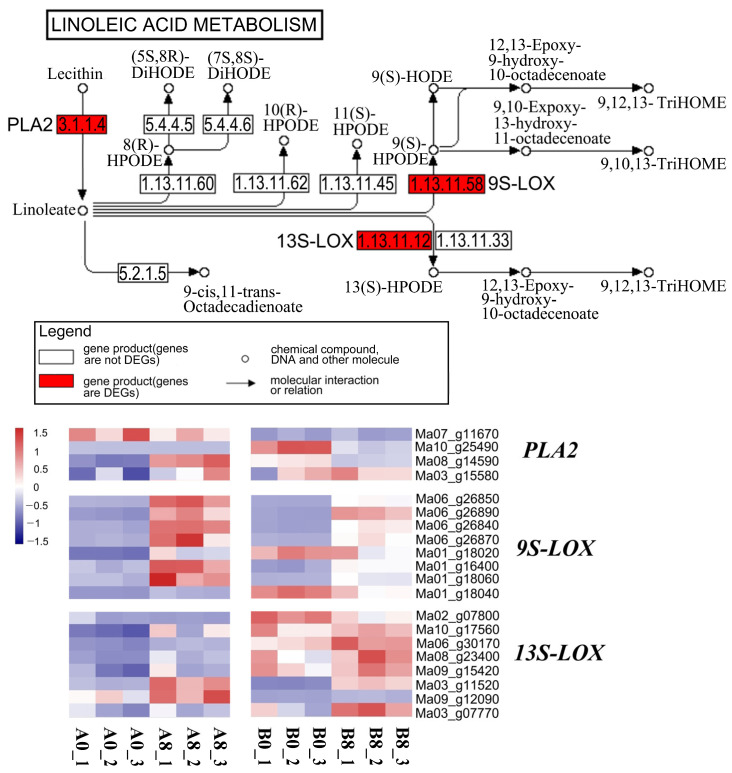
A schematic diagram of the “linoleic acid metabolism” pathway and the heatmap of DEGs involved in the pathway. The genes in “linoleic acid metabolism” were enriched in banana leaves and roots. Red indicates high expression, and blue indicates low expression. A0, banana leaves exposed to MeJA for 0 h; A8, banana leaves exposed to MeJA for 8 h; B0, banana roots exposed to MeJA for 0 h; B8, banana roots exposed to MeJA for 8 h.

**Figure 8 plants-13-00712-f008:**
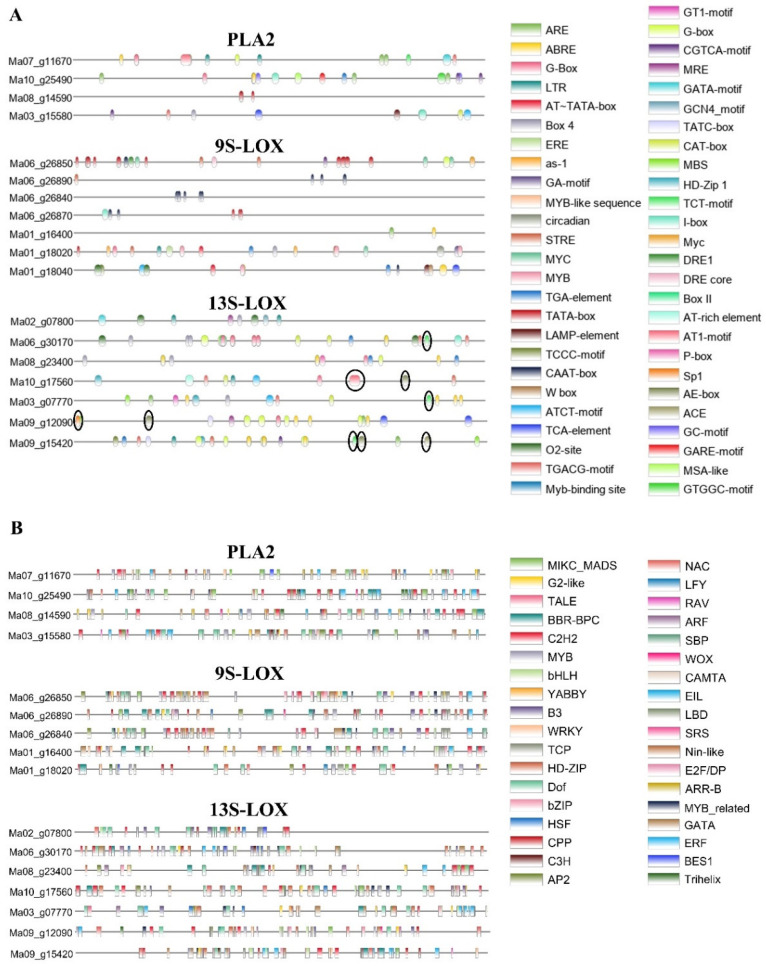
Analysis of the promoter region of the genes involved in linoleic acid metabolism. A 2000 bp-long sequence upstream of the CDS of the genes was obtained from the banana A genome data. (**A**) *Cis*-acting elements in gene promoters. Specific light-response elements in *13S-LOX* genes are circled in black. (**B**) Transcription factor (TF)-binding motifs.

**Figure 9 plants-13-00712-f009:**
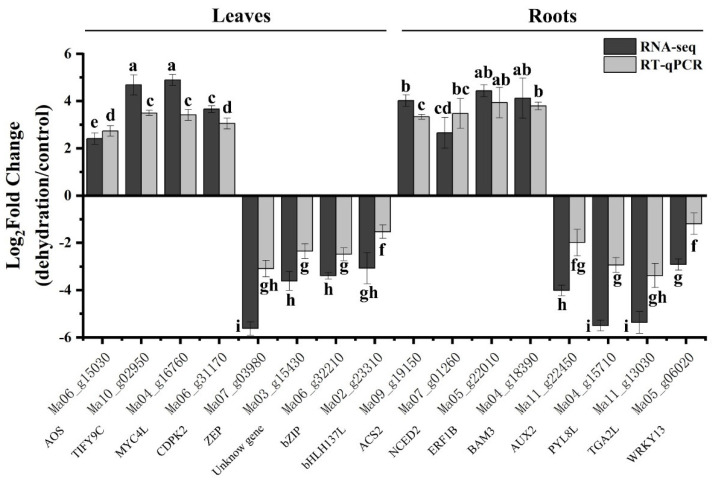
Verification of RNA-seq data using RT-qPCR. The expression patterns of DEGs in the leaves and roots of banana seedlings were validated using RT-qPCR. Actin was used as an internal control. Error bars indicate the standard error of the mean (±SE) of three replicate measurements. Different letters above the bars indicate significantly different values (*p* < 0.05), calculated using one-way analysis of variance (ANOVA) followed by Tukey’s multiple range test.

## Data Availability

Data are contained within the article and Appendix A.

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
