# Peer review of "Physiological and Transcriptomic Analyses Reveal the Mechanisms Underlying Methyl Jasmonate-Induced Mannitol Stress Resistance in Banana"

_plants, 2024, doi:10.3390/plants13050712_

Round 1

Reviewer 1 Report

Comments and Suggestions for Authors

Reviewer’s comments/suggestions on Plants-2724498 (Physiological and transcriptomic analysis reveals the mechanisms underlying MeJA-induced resistance in banana)

The article carries grammatical and syntax errors and requires thorough English correction. I recommend, the article not be accepted in its current format.

The design of the work carries serious flaws. The authors used four set of samples as, MeJA treated plants for 0h and 8h, and with 2days mannitol treatment for both. Thus, the samples are 0h-MeJA, 8h-MeJA, 0hMeJa+48hMa, 8hMeJa+48hMa. The authors did not clearly mention which samples were taken for RNA-seq analysis. The comparison of RNA-seq analysis for pooling out DEGs must be, 0MeJA vs 8MeJA, to identify the all the genes that are responsive to MeJA treatment. They should have compared this data with  0hMeJa vs 0hMeJa+48hMa, to pool out genes that are specifically regulated with Mannitol treatment. The this data should be compared with DEGs pooled out from 0hMeJa+48hMa vs 8hMeJa+48hMav, to get the genes that are MeJA responsive as well as with role in drought resistance. I find this as a flaw with designing of experiment.

The flow of article reminds of a lab discussion rather a journal material.  The figure legends does not carry sufficient information, (for eg. The authors did not even mention and explain about A0, B0, A8 and B8, in the figure legend of Fig. 2). The results and materials & methods and overlaps one another. The promoter was mentioned as reporter in materials and methods. The discussion volumes less. The references are not included in the text body of the article, and the supportive documents are missing. All these reflect the authors’ unpreparedness and a hurry in submitting the article.

 I recommend this article to be accepted in its current form.

Comments on the Quality of English Language

Comments added in the previous section

Author Response

Comments 1: The authors used four set of samples as, MeJA treated plants for 0h and 8h, and with 2days mannitol treatment for both. Thus, the samples are 0h-MeJA, 8h-MeJA, 0hMeJa+48hMa, 8hMeJa+48hMa.

Response 1: Thank you for pointing this out. We agree with this comment. Therefore, we have changed the names of group methods and applied them to the figure 1. And please see manuscript for details.

Line 92 – line 97: We changed the description of processing method.

Comments 2: The authors did not clearly mention which samples were taken for RNA-seq analysis.

Response 2: The description of the sample used for transcriptome sequencing had given in both the part of results and materials & methods. However, we made adjustments to avoid duplication.

Line 126-line 131, line 461-line 464.

Comments 3: The figure legends does not carry sufficient information, (for eg. The authors did not even mention and explain about A0, B0, A8 and B8, in the figure legend of Fig. 2).

Response 3: Thank you for helping us to notice that the legend was not comprehensive. We have added the legend for figures.

Line 119-line 123, line 152-line157, line199-line205, line218-line223, line 242-line 246, line 276-line 279, line296-line300, line319-line322, line339-line341

Comments 4: The results and materials & methods and overlaps one another.

Response 4: Agree, and we rewrite the method we treat plant material and made adjustment to other parts.

Line 453-line 459.

Comments 5: The promoter was mentioned as reporter in materials and methods.

Response 5: Agree. Thank you very much. We have fixed this mistake.

Line 490-line 492.

Comments 6: The discussion volumes less.

Response 6: Agree. We contributed to the discourse regarding the involvement of MeJA in enhancing plant resilience and elucidated the reaction of bananas towards adverse conditions.

Line 358- line 384

Comments 7: The references are not included in the text body of the article

Response 7: Agree. We have updated the references for the full article.

Comments 8: The supportive documents are missing.

Response 8: Thank you for your comments. We have included a supplementary document at the conclusion of the manuscript.

Comments 9: They should have compared this data with 0hMeJa vs 0hMeJa+48hMa, to pool out genes that are specifically regulated with Mannitol treatment. The this data should be compared with DEGs pooled out from 0hMeJa+48hMa vs 8hMeJa+48hMav, to get the genes that are MeJA responsive as well as with role in drought resistance.

Response 9:

The reviewers' design ideas were magnificent to achieve a more accurate identification of MeJA response genes which associated with Ma stress resistance. Our biochemical experiments have revealed that the application of MeJA has a significant impact on banana seedlings, even in the absence of Ma stress. Importantly, this effect is independent of Ma treatment. In order to comprehensively analyze the genes induced by MeJA expression, we aim to conduct RNA-seq analysis to investigate changes in gene expression in banana seedlings before and after MeJA treatment. Furthermore, our data demonstrates that 9S-LOXs and 13S-LOXs exhibit widespread responsiveness to MeJA and are uniquely expressed in banana roots and leaves. This conclusion was drawn through a comparative analysis of gene differences between the 0hMeJa and 8hMeJa experimental groups, as well as an examination of differential gene expression patterns in roots and leaves. These findings strongly support the notion that 9S-LOXs and 13S-LOXs play pivotal roles as key genes contributing to enhanced plant resistance mediated by MeJA. Thank you so much for the valuable suggestions. We truly appreciate them and will definitely incorporate these suggestions into our future research endeavors.

Response to Comments on the Quality of English Language

Point 1: The article carries grammatical and syntax errors and requires thorough English correction.

Response 1: We have conducted a comprehensive English revision of this article. The revised sections in the article have been highlighted but are currently hidden. You will be able to view all the revisions when reviewing it.

5. Additional clarifications

All revisions made to the article have been appropriately highlighted, while specific modifications have been concealed to ensure protection against any amendments that could affect the overall lines number of the manuscript.

Reviewer 2 Report

Comments and Suggestions for Authors

In the title, mention to mannitol stress in our title. 

The figures need to be improved and  

Improve the keywords.  under osmotic stress, cannot be use under as a keyword.

The references of introduction should be revised. 

The introduction section must be improved by adding more information on the sensitivity of banana to stress. 

Add more information on the biochemical change of 

The aim of the work must be rewritten. 

The resolution of figures must be improved. 

For Figure 9, add statistical analysis. 

In the discussion section, the references should be edited major errors has been found. 

The conclusion should be rewritten.

Why did the authors not include a control without addition of mannitol?

What is the number of plants used per replica? how many replicates per treatment?

 The references should be updated throughout the manuscript.

Comments on the Quality of English Language

Moderate English editing should be included. 

Author Response

Comments 1: In the title, mention to mannitol stress in our title.

Response 1: Many thanks for your suggestions, we had added the ‘mannitol stress’ in the title.

Line 3

Comments 2: The figures need to be improved. For Figure 9, add statistical analysis. The resolution of figures must be improved.

Response 2: Agree. Thank you for your suggestion. We have made adjustments to ensure that the figures are clearly visible, including resizing the words and enhancing the clarity of the figures.

Figure 3, Figure 4, Figure 5, Figure 6, Figure 7, Figure 9

Comments 3: Improve the keywords. under osmotic stress, cannot be use under as a keyword.

Response 3: Agree. We modified "Under osmotic stress" to "Mannitol-induced stress".

Line 30

Comments 4: The introduction section must be improved by adding more information on the sensitivity of banana to stress.

Response 4: We appreciate your advice. We have incorporated additional information regarding the sensitivity of bananas to stress in the introduction.

Line 37-line 49

Comments 5: Add more information on the biochemical change of the aim of the work must be rewritten.

Response 5: Thanks for your suggestion we have added a description of the aim for biochemical measurement.

Line 102-line 111

Comments 6: The references of introduction should be revised. In the discussion section, the references should be edited major errors has been found. The references should be updated throughout the manuscript.

Response 6: Agree. We have updated the references for the full article.

Comments 7: The conclusion should be rewritten.

Response 7: Thanks for your suggestion, we have revised the conclusion part.

Line 343-line 353

Comments 8: Why did the authors not include a control without addition of mannitol? What is the number of plants used per replica? how many replicates per treatment?

Response 8: We are currently unaware of the part which the control without additional mannitol. However, we have rewritten the treatment method of plant material mentioned in the article. We added the number of biological replicates in the experiment and the number of plants used in each replicate.

Line 92 -line 97; line 125-line 131; line 461-line 464

Additional clarifications

All revisions made to the article have been appropriately highlighted, while specific modifications have been concealed to ensure protection against any amendments that could affect the overall lines number of the manuscript.

Reviewer 3 Report

Comments and Suggestions for Authors

This study presents a study investigating the effects of exogenous methyl jasmonate (MeJA) on enhancing osmotic stress resistance in banana plants. The research showcases MeJA's positive impact on plant phenotype and antioxidant enzyme activity under mannitol-induced osmotic stress. Employing high-throughput RNA sequencing, the study delves into the molecular mechanisms behind MeJA-induced stress resistance in banana seedlings. The RNA-seq analysis reveals significant gene expression changes, highlighting specific pathways like linoleic acid metabolism. Notably, distinct gene expression patterns between leaves and roots shed light on tissue-specific responses to MeJA treatment. Identifying gene promoters and their responsive elements provides valuable insights into regulating MeJA-induced responses in banana seedlings. Overall, this study not only elucidates the mechanisms contributing to abiotic stress resilience in bananas but also offers potential avenues for enhancing banana varieties through molecular breeding strategies.

Major points:

1.       Line 168-169, Is 13,265 DEGs too much? As I learned about 36,000 genes expressed in bananas, 1/3 of genes are differentially expressed in roots. Do the authors have any explanations for that? I do suggest authors set a cutoff of the expression level (e.g., FPKM>=1) before performing further analysis.

2.       Libe 193, the same for the GO analysis with all the DEGs, are 4,140 GO terms too much? Besides, are there any different enriched terms for upregulated and downregulated genes?

Minor points:

1.       check the manuscripts including the formats (e.g., number format in the main text, fig.2, and Supp. Tables is not consistent), references, etc.

2.       Line 147, if 87% is the average percentage, please indicate it’s an average percentage.

3.       Line 164, what do authors mean by setting |log2 fold-change (FC)| > 0? Why did the authors set the parameter logFC without filtering any candidate genes?

4.       Fig 2B. “Differential Expressed Genes” -> “Differentially Expressed Genes”. Please label the genes without any differences (blue dots) in the figure.

Comments on the Quality of English Language

Need minor improvement

Author Response

Point-by-point response to Comments and Suggestions for Authors

Comments 1:

Line 168-169, Is 13,265 DEGs too much? As I learned about 36,000 genes expressed in bananas, 1/3 of genes are differentially expressed in roots. Do the authors have any explanations for that? I do suggest authors set a cutoff of the expression level (e.g., FPKM>=1) before performing further analysis.

Line 164, what do authors mean by setting |log2 fold-change (FC)| > 0? Why did the authors set the parameter logFC without filtering any candidate genes?

Response 1: Response to Reviewer 3, thank you so much for your valuable comments. We have observed an inadvertent error in setting the fold-change (FC) threshold for screening differentially expressed genes (DEGs), prompting us to revise the criteria to | log2 fold-change (FC)| > 1. Consequently, a reduction in the number of DEGs has been noted. Specifically, our revised analysis reveals that there are now 7,966 differentially expressed genes in the root, contrasting with the previously reported number of 13,256. These updated findings have been incorporated into both the part of results 2.3 and Figure 2 of this study.

Line 20-line 21, Line 159-line 170

Comments 2:  Libe 193, the same for the GO analysis with all the DEGs, are 4,140 GO terms too much? Besides, are there any different enriched terms for upregulated and downregulated genes?

Response 2: I agree. Thanks for your inquiries and suggestions regarding the GO enrichment analysis results section. Based on your comments and the most recent findings from DEGs screening, we have revised and updated both the GO enrichment results and Fig. 3 accordingly. The enriched terms of upregulated and downregulated genes are included there.

Line 172-line 220

Comments 3:  Line 147, if 87% is the average percentage, please indicate it’s an average percentage.

Response 3: Thank you for your inquiry, which has brought to our attention that the provided description lacks clarity. It should be noted that the mentioned 87% represents the minimum percentage rather than an average. However, we have revised the data to reflect an average percentage and ensured its explicit clarification.

Line 142-line 143

Comments 4:  Fig 2B. “Differential Expressed Genes” -> “Differentially Expressed Genes”. Please label the genes without any differences (blue dots) in the figure.

Response 4: Thanks for your comments. We changed the DEGs filters and updated the results in Fig. 2B, C. And we label the genes without any differences (gray dots).

Figure 2

Comments 5:  Check the manuscripts including the formats (e.g., number format in the main text, fig.2, and Supp. Tables is not consistent), references, etc.

Response 3: Thank you very much for your comments. We have reunified the text format and size in the manuscripts.

Response to Comments on the Quality of English Language

Point 1: Need minor improvement.

Response 1: We further improved the language and established a standardized writing format.

Additional clarifications

Based on the newly acquired differentially expressed genes (DEGs), we conducted a re-analysis of the KEGG pathways, resulting in slight modifications as depicted in Figure 4. In banana leaves, the pathways "photosynthesis-antenna proteins" (mus00196), "photosynthesis" (mus00195), and "linoleic acid metabolism" (mus00591) still exhibit the highest enrichment index. Similarly, in banana roots, the pathways with the highest enrichment index remain unchanged: "linoleic acid metabolism" (mus00591), "selenocompound metabolism" (mus00450), and "biotin metabolism" (mus00780). These pathways are still worth discussing and do not affect subsequent results.

Figure 4

Round 2

Reviewer 2 Report

Comments and Suggestions for Authors

 language edition should be done. 

Comments on the Quality of English Language

 language edition should be done. 

Author Response

Point 1: language edition should be done.

Response 1: We appreciate your comments. We further improved the language and established a standardized writing format.
